# Research of Water Molecules Cluster Structuring during *Haberlea rhodopensis* Friv. Hydration

**DOI:** 10.3390/plants11192655

**Published:** 2022-10-10

**Authors:** Ignat Ignatov, Fabio Huether, Nikolai Neshev, Yoana Kiselova-Kaneva, Teodora P. Popova, Ralitsa Bankova, Nedyalka Valcheva, Alexander I. Ignatov, Mariana Angelcheva, Ivan Angushev, Sadek Baiti

**Affiliations:** 1Scientific Research Center of Medical Biophysics (SRCMB), 1111 Sofia, Bulgaria; 2EVODROP AG, 8048 Zürich, Switzerland; 3Faculty of Physics, Sofia University “St. Kliment Ohridski”, 1000 Sofia, Bulgaria; 4Department of Biochemistry, Molecular Medicine and Nutrigenomics, Medical University-Varna, 9002 Varna, Bulgaria; 5Faculty of Veterinary Medicine, University of Forestry, 10 Kl. Ohridski Blvd., 1756 Sofia, Bulgaria; 6Department of Internal Noncommunicable Diseases, Pathology and Pharmacology, Faculty of Veterinary Medicine, University of Forestry, 10 Kl. Ohridski Blvd., 1756 Sofia, Bulgaria; 7Faculty of Agriculture, Department Biochemistry, Microbiology, Physics, Trakia University, 6000 Stara Zagora, Bulgaria; 8Department of Kinesitherapy and Rehabilitation, National Sports Academy “B. Levski”, 1700 Sofia, Bulgaria; 9Medical University-Sofia, 1606 Sofia, Bulgaria; 10Nature Bioresearch, 35300 Fougeres, France

**Keywords:** EVOdrop device, hydrogen-rich water, NES, DNES, *Haberlea rhodopensis* Friv.

## Abstract

*Gesneriaceae* plant family is comprised of resurrection species, namely *Boea hygrometrica* and *Paraboea rufescens*, that are native to the Southeast Asia and *Haberlea rhodopensis*, *Ramonda myconi*, and *Ramonda serbica*, which are mainly found in the Balkan Peninsula. *Haberlea rhodopensis* is known to be able to survive extreme and prolonged dehydration. Study was carried out after the dried plant *Haberlea rhodopensis* Friv. had been hydrated and had reached its fresh state. Two juice samples were collected from the plant blossom: The first sample was prepared with 1% filtered water through a patented EVOdrop device. Then the sample was saturated with hydrogen with EVOdrop booster to a concentration of 1.2 ppm, pH = 7.3, ORP = −390 mV. This first sample was prepared with filtered tap water from Sofia, Bulgaria. The second sample, which was a control one, was developed with tap water from Sofia, Bulgaria, consisting of 1% solutions of *Haberlea rhodopensis*. A study revealed that during the drying process in *H. rhodopensis* the number of free water molecules decreases, and water dimers are formed. The aim of our study was to determine the number of water molecules in clusters in 1% solutions of hydrated *H. rhodopensis* plants. Results were analyzed according to the two types of water used in the experiment. Th EVOdrop device is equipped with an ultranano membrane and rotating jet nozzle to create a vortex water and saturation thanks to a second device EVObooster to obtain hydrogen-rich water. In the current study Hydrogen-rich water is referred to as Hydrogen EVOdrop Water (HEW). Research was conducted using the following methods—spectral methods non-equilibrium energy spectrum (NES) and differential non-equilibrium energy spectrum (DNES), mathematical models, and study of the distribution of water molecules in water clusters. In a licensed Eurotest Laboratory, the research of tap water before and after flowing through the EVOdrop device was proven. Studies have been carried out on the structuring of water molecule clusters after change of hydrogen bond energies. The restructuring comes with rearrangement of water molecules by the energy levels of hydrogen bonds. Local extrema can be observed in the spectrum with largest amount of water molecules. The structural changes were tested using the NES and DNES spectral methods. The conducted research proved that the application of EVOdrop device and EVObooster changes the parameters of water to benefit hydration and health.

## 1. Introduction

*Haberlea rhodopensis* Friv. is a Balkan endemic plant primarily found in the Rhodope Mountains in Bulgaria. It is known for its drought resistance and biosis–anabiosis–biosis cycle [1]. It is a protected species under the Law on Biological Diversity in Bulgaria [2] The plant is listed in the Red Book of Bulgaria in the category of rare species, Balkan endemic [3]. The biosis–anabiosis–biosis cycle may unveil new properties of the water. A study revealed that, during the drying process in *H. rhodopensis*, the number of free water molecules decreases, and water dimers are formed (Kuroki, et al., 2019) [4]. After contact with water, *H. rhodopensis* plants straighten up. However, the reasons for this phenomenon are not fully known. We consider it to be possible to evaluate hydration with NES and DNES methods and based on water properties when applied after dehydration of *H. rhodopensis*.

The equipment used for these studies is the EVOdrop device with an ultranano membrane [5] and rotating jet nozzle for vortex water [6]. The ultranano membrane is a competitor to the reverse osmosis membrane. The rotating jet nozzle for vortex water [7] is equipped with three injection nozzles according to the golden ratio and algorithm. Some research has been conducted on the effects of magnetic field on water [8,9,10] and nanomembranes [11,12].

Hydroxyl groups (-OH) in H_2_O molecules are polar. A covalent bond of water molecules is a chemical bond that involves the sharing of electron pairs between O and H atoms. Between H_2_O molecules, there are electromagnetic hydrogen bonds Hydrogen bonds are weaker than covalent bonds. The water molecules could be bonded into complex intermolecular clusters, described by a general formula (H_2_O)_n_. The NES and DNES spectral methods are related to the research of parameters of hydrogen bonds, with estimation of the effects in a 1% solution (*v*/*v*) *H. rhodopensis* in filtered water, prepared with a patented EVOdrop device and saturated with hydrogen with an EVOdrop booster. As a control, 1% *H. rhodopensis* with tap water was used.

Research by Smith et al. and Keutsch and Saykally showed water clusters with 3 to 50 water molecules [13,14]. Different water cluster models are also described in the investigations conducted by Fowler et al. [15], Shu et al. [16], Chaplin [17], Sykes [18], Liu, Cruzan and Saykally [19], Choi and Jordan [20], Loboda and Goncharuk [21] and Timothy and Zwier [22].

The following methods have been generally used to study water clusters—^1^H NMR [23,24], far-infrared [25], vibration–rotation–tunneling (VRT) [19], neutron diffraction [26], and the SCC-DFTB Method [18,27]. A cluster model at (E = −0.1387 eV) (λ = 8.95 μm) (ṽ = 1117 cm^−1^) has been proposed with 20 water molecules in a dodecahedral structure with diameter of the circumscribed sphere equal to 0.822 nm [28,29,30]. The basis of this research is the NES and DNES methods of Antonov et al. [31,32,33].

It is accepted that the aqueous solutions can undergo autoprotolysis, i.e., the H^+^ proton is released from the H_2_O molecule and then transferred and accepted by the neighboring H_2_O molecule, resulting in the formation of hydronium ions as H_3_O^+^, H_5_O_2_^+^, H_7_O_3_^+^, H_9_O_4_^+^, etc. Thus, water should be considered as an associated liquid composed of a set of individual H_2_O molecules, linked together by hydrogen bonds and weak intermolecular van der Waals forces [19]. The simplest example of such a complex can be a water dimer.

The research has shown that 1% solution of *H. rhodopensis* in water filtered by EVOdrop with hydrogen has the highest hydrogen bonds energies among water molecules at (−0.1112 eV; 11.3 μm; 887 cm^−1^). The EVOdrop device has an nano membrane and rotating jet nozzle for vortex water and saturation, thanks to a second device EVObooster for hydrogen-rich water. There is an increase in the local extremum in the spectrum compared to the control sample with 1% solution of *H. rhodopensis* in tap water filtered by EVOdrop with hydrogen.

The local extremum at (−0.1112 eV; 11.3 μm; 887 cm^−1^) is related to calcium conductivity [32,34]. Some studies reveal that signaling-related genes encoding a calcium channel protein are activated after hydration of *H. rhodopensis* [34]. Calcium is involved in the regulation of DT mechanisms [35]. The calcium ions in xyloglucan enhance strength and flexibility [36].

The aim of the study was to show that water filtered by EVOdrop with hydrogen has better hydration properties and structuring of water molecules into clusters.

The excepted applications are for positive effects on human health.

## 2. Results and Discussion

The reported results are average values between the results of the application of the device to test 10 different water samples after treatment with the EVOdrop devices and 10 control water samples. For each sample, 10 measurements were performed. There was a statistically significant difference between the results of the two groups of samples and the control samples according to Student’s *t* test with *p* < 0.05.

### 2.1. Mathematical Models of Clusters in a 1% Solution of Haberlea rhodopensis Friv. Blossom Extract

A mathematical model of the number of water molecules [37,38,39] according to the energy of hydrogen bonds in a 1% solution of *H. rhodopensis* blossom extract has been developed (Table 1; Figure 1).

The distribution of the number of water molecules in a 1% solution of *H. rhodopensis* blossom extract in EVOdrop-filtered, hydrogen-saturated tap water and in the control sample of tap water from Sofia, Bulgaria in accordance with the energy of hydrogen bonds is presented in Table 1.

Figure 1 presents the distribution of the number of water (H_2_O) molecules in EVOdrop filtered tap water and saturated with Hydrogen (H_2_) (sample) and tap water (control sample) according to the energy of hydrogen bonds. The model shows the number of water molecules and their structuring in clusters.

The function f(E) is a distribution spectrum according to energies. The non-equilibrium energy spectrum (NES) is measured in eV^−1^.

The local extremum at E = −0.11 eV; λ = 11.3 μm (ṽ = 887 cm^−1^) is specific to calcium carbonate ions [32,40]. The local extremum at E = −0.1112 eV or (λ = 11.3 μm; ṽ = 887 cm^−1^) is typical for calcium conductivity [41,42].

In 2016, Kostainova and co-authors performed in vitro research on keratinocytes. They found that *H. rhodopensis* extracts affect the cell periphery of these cells. The keratinocytes were cultured under standard conditions and supplemented with additional calcium ions (Ca^2+^) in order to stimulate tight junction formation, thereby suppressing proliferative activity [43].

### 2.2. Results from Spectral Analysis of EVOdrop Water with NES and DNES Methods

Measurements with the NES and DNES spectral methods show a significant difference between EVOdrop water and the control sample.

The test sample consisted of 1% *H. rhodopensis* blossom extract in filtered water, obtained with the patented EVOdrop device and saturated with hydrogen with the EVOdrop booster. The control sample was of 1%. *H. rhodopensis* blossom extract in deionized water.

The result for the test sample in the NES-spectrum was −0.1203 eV, while for the control sample it was −0.1151 eV. The values of ∆E for EVOdrop water, measured by the DNES method, were in the interval (−0.0052 eV). The highest number of water clusters in the sample was 15 (−0.1112 eV; 11.3 μm; 887 cm^−1^). The number of water molecules in the control sample was 2.

The difference is considerable and shows a higher-level structuring of water clusters in comparison with the control sample. Recently, Ignatov et al. [39] found that in water, clusters with different numbers of H_2_O molecules are formed due to the formation of hydrogen bonds. The average hydrogen bonds energy (HBE) increases with the number of H_2_O molecules in the clusters, and with the evaporation of water droplets. According Mehandjiev et al. [44], in bulk water, at the beginning of evaporation, the maximum number of clusters has an average HBE of (−E) = 0.1162 eV and contains 12–13 H_2_O molecules. Discrete changes in HBE of water clusters have the same value and are based on the formation of clusters with different numbers of water molecules.

*H. rhodopensis*, tested in our research, is a “resuscitating” plant with unique properties, a Bulgarian endemite. This is the only plant that recovers after a long drying time and has an ability to survive up to 31 months dried in an herbarium [44]. It is the only one in which Kuroki et al. [6] proved the presence of clusters of two water molecules in a dry state. There are no proven cluster formations in other plants. When *H. rhodopensis* dries out, dimers of water molecules are formed [6]. When being watered the dry plant turns green. After dehydration of *H. rhodopensis*, the degree of hydration can be estimated with the NES and DNES methods depending on the water quality.

Our results with *H. rhodopensis* show that after drying, the formation of clusters of water molecules depends on the type of water. The experiments were conducted with deionized and filtered EVOdrop water. The EVOdrop device enables the formation of clusters in water. The structuring of the water molecules was established, which are analyzed with NES and DNES methods.

In recent years, *H. rhodopensis* has drawn the attention of researchers due to its beneficial effects on human and animal health. The extract of the plant contains high levels of flavonoid antioxidants [45,46]. We used such an extract in the present study. The plant has a tonic and anti-aging effect. In folk medicine, it is used to cleanse the stomach, liver, kidneys, and blood vessels [47,48]. Aqueous and alcoholic extracts of *H. rhodopensis* have shown unique medical and pharmaceutical potential, related to their antioxidant, radioprotective, antimicrobial, antimutagenic, immunostimulatory, anticancer, and anti-aging effects. The extract could be used in phytotherapy, human and veterinary medicine and cosmetics [49,50]. Our results show that its unique useful properties can be enhanced using EVOdrop technology.

## 3. Materials and Methods

### 3.1. Plant H. rhodopensis Friv.

Ethanol extract from the *H. rhodopensis* leaves and blossom were used in our study (Figure 2). *H. rhodopensis* Friv. oil has the following chemical composition (Table 2) [51]:

Figure 3 illustrates flowers of *H. rhodopensis*.

### 3.2. The EVOdrop Turbine Water Purifier

The proprietary operating principle and developed geometry of the EVOdrop turbine (Figure 4) allow for highly efficient treatment. Incoming water passes through the rotating turbine, driving it with its pressure, which in turn makes water pass through the rotating device. Specific outcomes of this treatment are based on magnetohydrodynamic forces [4,5]. Figure 4 shows EVOdrop’s turbine operation principle.

Figure 5 illustrates the EVOdrop filter.

### 3.3. The EVOdrop Booster for Hydrogen-Rich Water

The EVOdrop Booster produces hydrogen-rich water. EVOdrop hydrogen water has a concentration of hydrogen (Figure 6).

Figure 7 illustrates the EVObooster device for EVOdrop hydrogen water.

The biological effects of hydrogen-rich water at a concentration of 0.08–1.5 ppm are described [17,18]. In the current study, hydrogen-rich water is referred to as Hydrogen EVOdrop Water (HEW).

### 3.4. Differential Non-Equilibrium Energy Spectrum (NES) and Differential Non-Equilibrium Energy Spectrum Spectral Analyses (DNES)

The device of the author A. Antonov [52,53,54,55] for spectral analysis with NES and DNES methods is based on an optical principle. The evaporation of water drops is performed in a hermetic camera with a glass plate and water-proof transparent pad which consists of thin maylar folio. Evaporation of water drops was performed at a stable temperature of 22 °C. The drops were placed on a BoPET (biaxially oriented polyethylene terephthalate) foil with a 350 µm thicknesses (Figure 8).

The parameters are as follows:Monochromatic filter with wavelength λ = 580 ± 7 nm;Angle of evaporation of water drops from 72.3° to 0°;Energy range of hydrogen bonds among water molecules is λ = 8.9–13.8 µm or E = −0.08–−0.1387 eV.

The energy (E) of hydrogen bonds among H_2_O molecules in the water sample is measured in eV. The function f(E) is called the energy distribution spectrum. The energy spectrum of water is characterized by a non-equilibrium process of water droplet evaporation; this is non-equilibrium energy spectrum (NES) and is measured in eV^−1^. DNES is defined as the difference
∆f(E) = f (samples of water) *−* f (control sample of water),

DNES is measured in eV^−1^

where f(*) denotes the evaluated energy [31,32].

### 3.5. Filtration with EVOdrop Filter for Tap Water, Sofia, Bulgaria

Table 3 illustrates the physicochemical parameters of tap water from Sofia, Bulgaria before and after filtration with the EVOdrop device. The certificate No. 10216/21.07.2022 corresponds to the water before filtration, and No. 10217/14.07.2022 after filtration, with the EVOdrop device. The research was performed according to the parameters of Ordinance No. 9/2001, Official State Gazette, issue 30, and Decree No. 178/23.07.2004 regarding the quality of water intended for consumption and domestic uses in the accredited laboratory “Eurotest control” JSC, Sofia, Bulgaria [56].

The research of physicochemical filtration with the EVOdrop filter shows that there is filtration of molecules with bigger sizes, such as hydrocarbonate ions (HCO_3_^−^). The result for HCO_3_^−^ is from 27.5 ± 2.8 to less than 24.4 mg·L^−1^. There is an increase in pH from acidic value to alkaline—from 6.73 ± 0.11 to 8.88 ± 0.11. The hardness of tap water after filtration is reduced from 1.76 ± 0.5 to 0.98 ± 0.24.

## 4. Conclusions

The unique properties of *Haberlea rhodopensis* Friv., related to interaction between the water molecules during the biosis–anabiosis–biosis cycle, were used to investigate EVOdrop nanofiltration technology of tap water and its enrichment with hydrogen in terms of hydration and water molecules clustering. The statistically significant results clearly demonstrated that EVOdrop treatment leads to a shift in the hydrogen bonds energy distribution towards larger values along with the corresponding formation of local maxima. Based on previous results, health benefits of such water treatment can be expected in the areas of malignant growth inhibition and tissue regeneration for humans.

## Figures and Tables

**Figure 1 plants-11-02655-f001:**
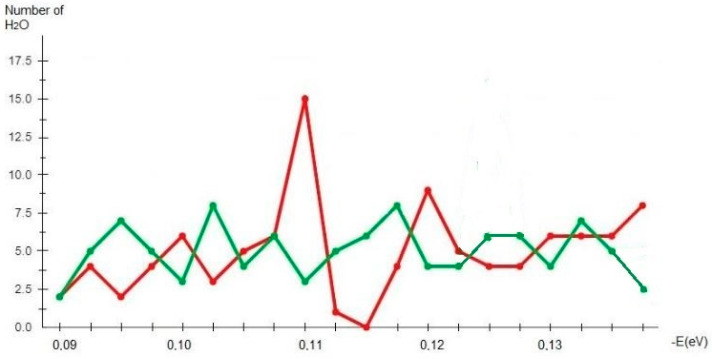
Distribution of the number of water (H_2_O) molecules in EVOdrop water and tap water according to the energy of hydrogen bonds.

**Figure 2 plants-11-02655-f002:**
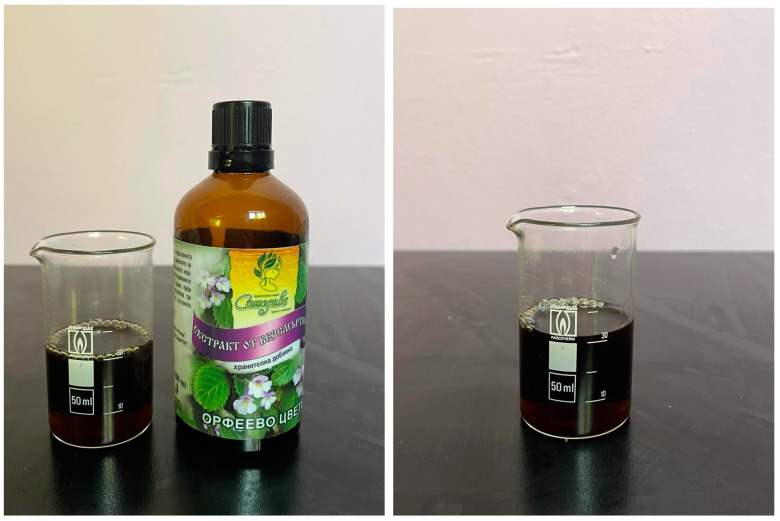
*Haberlea rhodopensis* blossom extract.

**Figure 3 plants-11-02655-f003:**
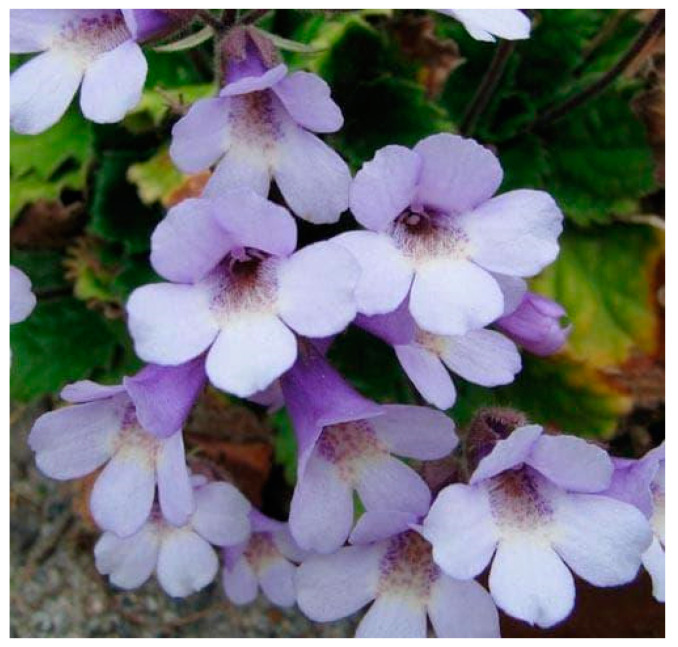
Flowers of *Haberlea rhodopensis*.

**Figure 4 plants-11-02655-f004:**
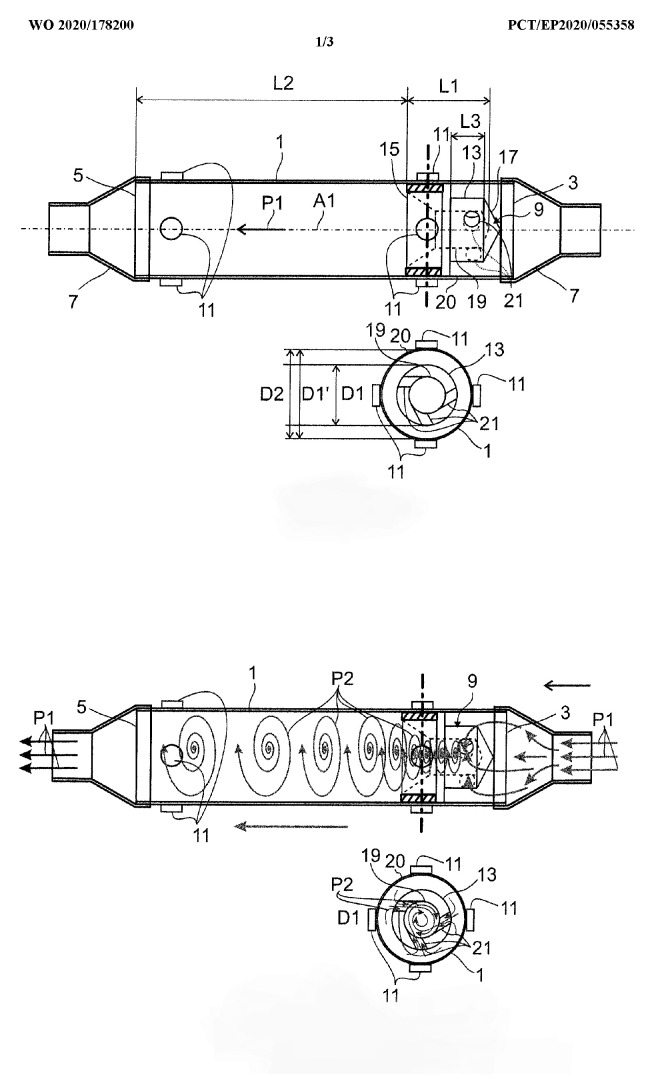
The EVOdrop turbine operation principle.

**Figure 5 plants-11-02655-f005:**
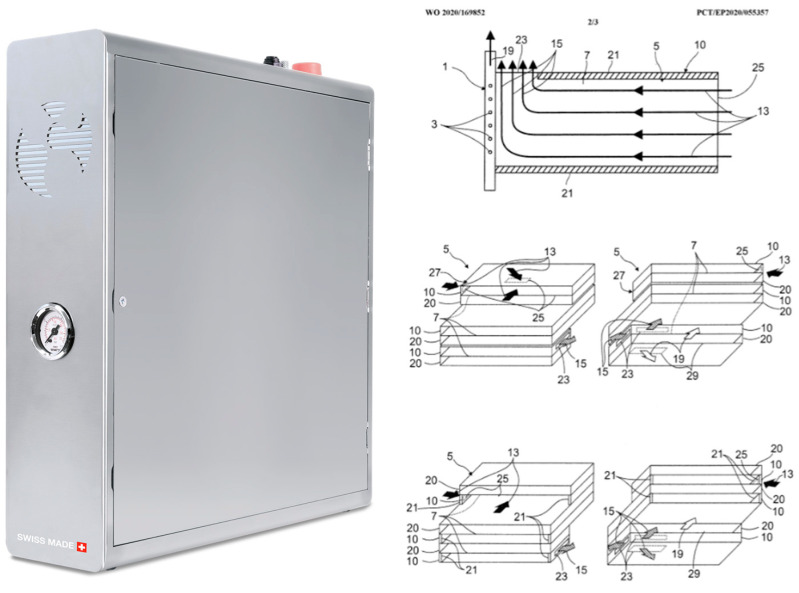
The EVOdrop filter.

**Figure 6 plants-11-02655-f006:**
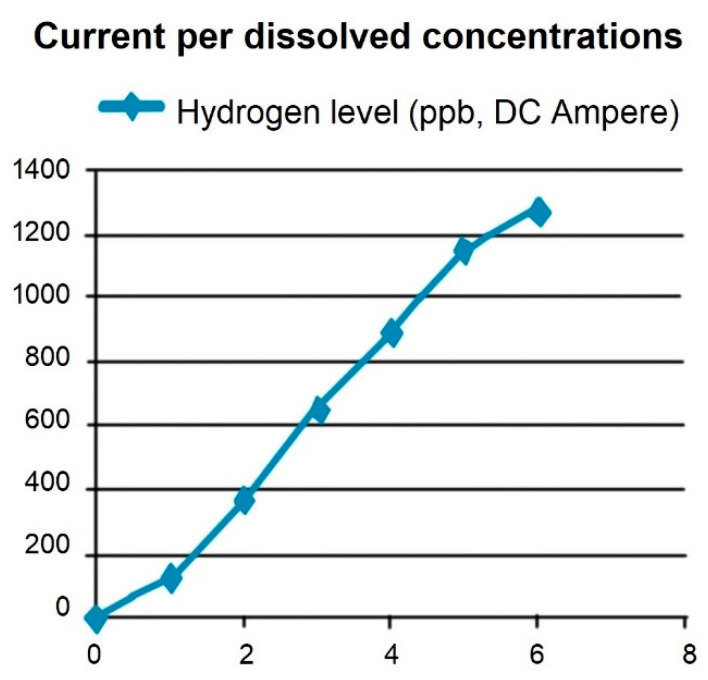
Hydrogen (H₂) concentration of EVOdrop hydrogen water (1.2 ppm). Correspondence of hydrogen concentration (ppb) of EVOdrop hydrogen water and the DC Current (Ampers).

**Figure 7 plants-11-02655-f007:**
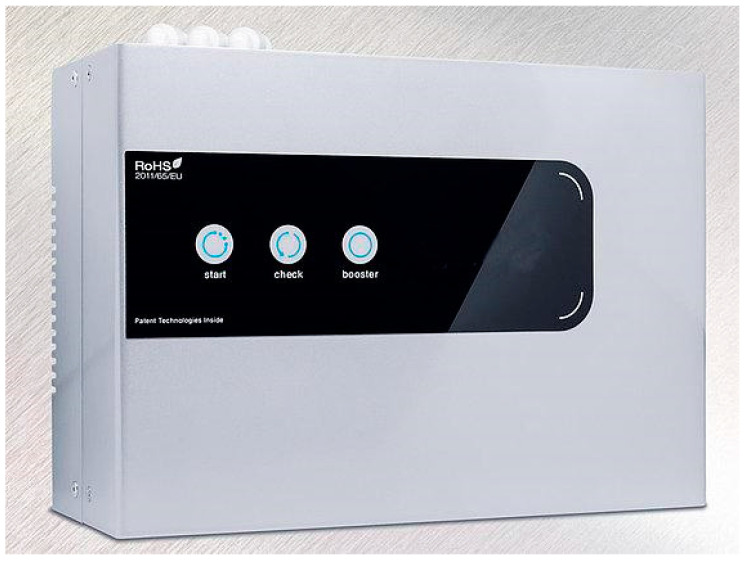
The EVObooster device for EVOdrop hydrogen water.

**Figure 8 plants-11-02655-f008:**
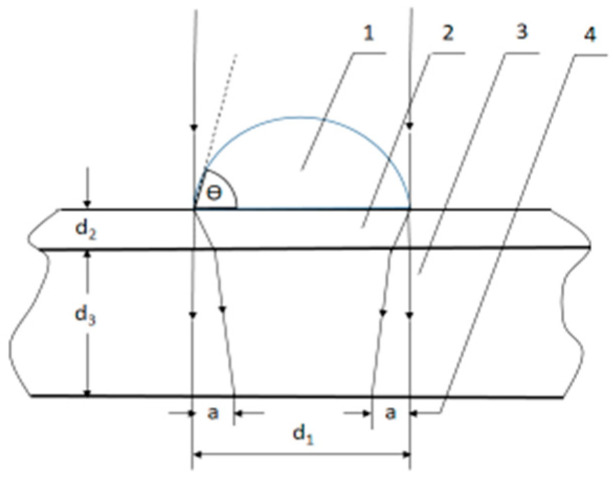
Operating principle of the method for measurement of wetting angle of liquid drops on a hard surface: 1—drop, 2—thin maylar foil, 3—glass plate, 4—refraction ring width. The wetting angle θ is a function of a and d_1_.

**Table 1 plants-11-02655-t001:** Distribution of the number of water (H_2_O) molecules in a 1% solution of *Haberlea rhodopensis* Friv. blossom extract in EVOdrop-filtered tap water saturated with Hydrogen (H_2_) and in the control sample of tap water from Sofia, Bulgaria according to the energy of hydrogen bonds.

−E(eV) x-Axis	Number of Water Molecules		−E(eV) x-Axis	Number of Water Molecules	
	1% Solution *H. rhodopensis* (Sample)	1% Solution *H. rhodopensis* (Control Sample)		1% Solution *H. rhodopensis* (Sample)	1% Solution *H. rhodopensis* (Control Sample)
0.0912	2	2	0.1162	0	6
0.0937	4	5	0.1187	4	8
0.0962	2	7	0.1212	9 ^2^	4 ^2^
0.0987	4	5	0.1237	5	4
0.1012	6	3	0.1262	4	6
0.1037	3	8	0.1287	4	6
0.1062	5	4	0.1312	6	4
0.1087	6	6	0.1337	6	7
0.1112	15 ^1^	3 ^1^	0.1362	6	5
0.1137	1	5	0.1387	8 ^3^	2 ^3^

Notes: ^1^
^1^E = −0.1112 eV or (λ = 11.3 μm; ṽ = 887 cm^−1^) is the local extremum for calcium conductivity. ^2^
^1^E = −0.1212 eV or (λ = 10.23 μm; ṽ = 978 cm^−1^) is the local extremum for anti-inflammatory effects. ^3^
^1^E = −0.1387 eV or (λ = 8.95 μm; ṽ = 1119 cm^−1^) is the local extremum for inhibition of development of tumor cells.

**Table 2 plants-11-02655-t002:** Chemical composition of *H. rhodopensis* oil.

Compounds	μg·g^−1^ DW
flavonoids	
Luteolin	2730.18
Hesperidin	928.56
Kaempferol	578.52
Phenolic Acids	
Ferulic acid	630.48
Sinapic acid	580.80

**Table 3 plants-11-02655-t003:** Physicochemical parameters of the tap water from Germany before and after filtration with EVOdrop device.

Controlled Parameter	Measuring Unit	Maximum Limit Value	Before EVOdrop	After EVOdrop
1. pH	pH values	≥6.5 and ≤9.5	6.73 ± 0.11	8.88 ± 0.11
2. Total hardness	mgekv·L^−1^	12	1.76 ± 0.5	0.98 ± 0.24
3. Calcium (Ca^2+^)	mg·L^−1^	150	12.7 ± 1.3	12.7 ± 1.3
4. Magnesium (Mg^2+^)	mg·L^−1^	80	21.2 ± 2.1	4.2 ± 0.4
5. Hydrocarbonates (HCO_3_^−^)	mg·L^−1^	-	27.5 ± 2.8	<24.4
6. Carbonates (CO_3_^2−^)	mg·L^−1^	-	<12	<12
7. Sodium (Na^+^)	mg·L^−1^	200	5.7 ± 0.9	5.7 ± 0.9
8. Potasium (K^+^)	mg·L^−1^	-	1.7 ± 0.2	1.6 ± 0.2
9. Manganese	µg·L^−1^	50	3.8 ± 0.4	3.6 ± 0.4
10. Zinc	mg·L^−1^	4	0.074 ± 0.07	0.02 ± 0.002

## Data Availability

Not applicable.

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
