# Peer review of "Research of Water Molecules Cluster Structuring during Haberlea rhodopensis Friv. Hydration"

_plants, 2022, doi:10.3390/plants11192655_

Round 1

Reviewer 1 Report

The manuscript is well written, clearly structured and provides interesting results. It introduces the import of EVOdrop treatment which leads to a shift in the hydrogen bonds energy distribution towards larger values along with the corresponding formation of local maxima. I recommend authors to critically check the manuscript for English improvement to avoid typos.  Further applications of EVOdrop 265 nano-filtration technology should be discussed.

Reviewer 2 Report

 The authors can improve the introduction with additional publications about Rhodope haberlea. What other plants in the world have similar properties?

Reviewer 3 Report

The authors can show 3-4 additional references concerning the properties of Rhodope haberlea.

The sentence on lines 134-135 is the same as that on lines 140-141, and I recommend that the sentence on lines 134-135 be deleted. 

In the legend of Figure 6, the abscissa parameter should be indicated, not just the ordinate parameter. I suggest, for example: Correspondence of Hydrogen concentration (ppb) of EVOdrop hydrogen water and the DC Current (Ampers).